# Integrating Metabolomics and Gene Expression Underlying Potential Biomarkers Compounds Associated with Antioxidant Activity in Southern Grape Seeds

**DOI:** 10.3390/metabo13020210

**Published:** 2023-01-31

**Authors:** Ahmed G. Darwish, Md Moniruzzaman, Violeta Tsolova, Islam El-Sharkawy

**Affiliations:** 1Center for Viticulture and Small Fruit Research, College of Agriculture and Food Sciences, Florida A&M University, Tallahassee, FL 32308, USA; 2Department of Biochemistry, Faculty of Agriculture, Minia University, Minia 61519, Egypt

**Keywords:** antioxidant, metabolome, genes, muscadine genotypes, seed, biomarker compounds

## Abstract

Different southern grape (Muscadine) genotypes (*Muscadinia rotundifolia* Michx.) were evaluated for their contents of metabolites in ripe berries. The metabolome study identified 331 metabolites in ripening skin and seed tissues. The major chemical groups were organic acids, fatty acyls, polyketides, and organic heterocycle compounds. The metabolic pathways of the identified metabolite were mainly arginine biosynthesis, D-glutamine, D-glutamate metabolism, alanine, aspartate metabolism, aminoacyl-tRNA biosynthesis, and citrate cycle. Principal component analysis indicated that catechin, gallic acid, and epicatechin-3-gallate were the main metabolites existing in muscadine seed extracts. However, citramalic and malic acids were the main metabolites contributing to muscadine skin extracts. Partial least-squares discriminant analysis (VIP > 1) described 25 key compounds indicating the metabolome in muscadine tissues (skin and seed). Correlation analysis among the 25 compounds and oxidation inhibition activities identified five biomarker compounds that were associated with antioxidant activity. Catechin, gallic acid, epicatechin-3-gallate, fertaric acid, and procyanidin B1 were highly associated with DPPH, FRAP, CUPRAC, and ABTS. The five biomarker compounds were significantly accumulated in the seed relative to the skin tissues. An evaluation of 15 antioxidant-related genes represented by the 3-dehydroquinate dehydratase (*DHD*), shikimate kinase (*SK*), chalcone synthase (*CHS*), anthocyanidin reductase (*ANR*), laccase (*LAC*), phenylalanine ammonia-lyase (*PAL*), dihydroflavonol 4-reductase (*DFR*), 3-dehydroquinate synthase (*DHQS*), chorismate mutase (*CM*), flavanone-3-hydroxylase (*F3H*), cinnamoyl-CoA reductase (*CCR*), cinnamyl alcohol dehydrogenase (*CAD*), leucoanthocyanidin reductase (*LAR*), gallate 1-β-glucosyltransferase (*UGT*), and anthocyanidin 3-O-glucosyltransferase (*UFGT*) encode critical enzymes related to polyphenolics pathway throughout four developmental stages (fruit-set FS, véraison V, ripe-skin R, and ripe-seed; S) in the C5 genotype demonstrated the dramatic accumulation of all transcripts in seed tissue or a developmental stage-dependent manner. Our findings suggested that muscadine grape seeds contain essential metabolites that could attract the attention of those interested in the pharmaceutical sector and the plant breeders to develop new varieties with high nutraceutical value.

## 1. Introduction

The grape seed extract is a rich source of antioxidants. It has been linked to a wide range of beneficial health qualities against several diseases, such as inflammation, cardiovascular disease, hypertension, diabetes, cancer, peptic ulcer, microbial infections, high cholesterol, atherosclerosis, macular degeneration, poor circulation, and nerve damage [1,2,3,4]. Muscadines are a grape species that are well adapted to the southeastern climate of the United States. Their polyphenols have received increased attention due to their high content and potential bioactivity against numerous diseases [5,6,7,8,9,10,11]. They are reported to contain free and conjugated forms of hydroxybenzoic acids, ellagic acid, resveratrol, and flavonoids, including anthocyanins, quercetin, myricetin, kaempferol, ellagic acid, proanthocyanidins, and catechins, which are considered the most critical compounds coordinating antioxidant activity [12,13,14,15,16,17,18,19,20]. Moreover, muscadine grape extracts contain many other phytochemicals, such as troxerutin, fertaric acid, dopamine, and naringenin, exhibiting substantial antioxidant properties due to their ability to scavenge hydroxyl, peroxyl, and superoxide radicals [21,22,23,24,25,26,27]. Grape polyphenolics play an essential role in protecting endothelial cells from oxidative destruction and decreasing the inactivation of NO˙ (nitric oxide) radicals via the modulation of oxidation enzymes, including NADPH oxidase, superoxide dismutase, and glutathione peroxidase [28,29,30,31,32,33,34,35,36]. Despite the unique biochemical composition of muscadine grapes, the health benefits of nutritional supplements of muscadine seeds or other muscadine-derived food supplements as nutraceuticals or functional foods need more investigation. Therefore, besides using muscadine as a nutraceutical or cosmeceutical, it may be used as a traditional medicine or drug to treat diseases by developing it into other effective therapeutic formulations to improve imminent prospects.

The biosynthetic pathway of polyphenolic compounds involves several enzymes that catalyze sequential reactions [37]. They have considerable roles in plants related to growth, development, berry quality, and sensory attributes [38,39]. Phenolic content is highly influenced by genotypic variation, environment, organ type, and the stage of development [40]. In muscadine berries, polyphenolic compounds are accumulated mainly in the skin and seed. Flavonols and anthocyanins are berry skin’s most abundant phenolic compounds, whereas seeds are rich in flavan-3-ols and proanthocyanidins [40,41]. The phenols can be released more readily from berry skin than from seed, but their content is much lower than in the seeds [42].

The present work aimed to explore the metabolic profiles and antioxidant activities of phytochemicals accumulated in ripe muscadine berries, which could be categorized as biomarker compounds with antioxidant potential. The untargeted metabolome profiling strategy was used to identify phytochemical compounds underlying antioxidant activity in the ripe-skin (R) and -seed (S) of three muscadine genotypes. A multivariate analysis was also accomplished to illustrate biomarker-key compounds with antioxidant activity, which could be necessary for nutraceutical purposes. In addition, the transcription level of several genes encoding key enzymes that are involved in the polyphenolics biosynthetic pathway was assessed to confirm the genes that are involved in the metabolic reactions of the biomarker antioxidant compounds in muscadine during different developmental stages.

## 2. Materials and Methods

### 2.1. Chemicals

DPPH, Trolox, methanol (HPLC grade), acetic acid, TPTZ (2,4,6-Tri(2-pyridyl)-*s*-triazine), ABTS, SNP (silver nanoparticles), CuCl_2_, NCP (copper (II)-neocuproine), HoAc (glacial acetic acid), ammonium acetate buffer (AAB), and FeCl_3_ (ferric chloride) were purchased from Sigma-Aldrich Chemical Co. (St. Louis, MO, USA).

### 2.2. Muscadine Grape Materials 

There were three muscadine genotypes (*Muscadinia rotundifolia*), including the bronze cultivar ‘Late Fry (LF),’ and two-colored breeding lines, ‘C5-9-1 (C5)’ and ‘C6-10-1 (C6)’. All muscadine genotypes were grown at the Center for Viticulture and Small Fruit Research, Tallahassee, Florida (30°28′045.63″ N, 84°10′16.43″ W). The three genotypes were selected according to their diversity in DPPH activity at the stage of ripening (Figure 1A). Berry samples were collected in fruit-set (FS), véraison (V), and ripening stages. At the ripening stage (August–September 2020), the berries were collected and split into two tissues: ripe/skin (R) and ripe/seed (S). All samples were instantly preserved in LN2 (liquid nitrogen) and kept at −80 °C.

### 2.3. Muscadine Extracts

All frozen samples were ground to a fine powder using a Geno/Grinder 2010 (SPEX-SamplePrep 2010, Metuchen, NJ, USA). A total of 12 g and 3 g of skin and seed powder samples were extracted using 100 mL and 50 mL of methanol with a ratio (1:8; 1:16) to yield 5.40 g and 0.23 g, respectively. All extracts were kept on a shaker at 150 rpm/24 h in the dark/25 °C, followed by filtration using Whatman papers (41 ashless, Thomas Scientific, Swedesboro, NJ, USA). A rotaevaporator (Hei-Vap, Heidolph, Fisher Scientific, Waltham, MA, USA) was used to evaporate the remaining MeOH from all extracts at 40 °C and dried using a speed vacuum (Eppendorf AG 22331 Hamburg, Enfield, CT, USA). All extracts were kept at 4 °C in the dark for future analysis. A 10 mg/mL stock solution in DMSO of grape extracts was prepared to determine antioxidant activity.

### 2.4. Analysis of Antioxidant Activities

#### 2.4.1. DPPH Radical Scavenging Activity

DPPH radical scavenging activity was assessed as previously reported [43]. A serial dilution was prepared in a 96-well plate, obtaining final concentrations of 3.12, 12.5, 25, 50, and 100 μg/mL for each extract. A total of 100 μL of diluted samples were added to 100 μL of DPPH solution (200 μM) and then incubated in the dark for 30 min at 25 °C. A spectrowavelength (λ = 515 nm) was used to measure the absorbance using DMSO as a control. DPPH activity was assessed in triplicate of bio- and tech-replicates (*n* = 9). The results were expressed as DPPH inhibition %. The calibration curve was established using the inhibition rate of the Trolox solution.

#### 2.4.2. Ferric Reducing Antioxidant Potential (FRAP) Assay

This was undertaken following the previous method [44]. A total of 280 µL freshly prepared FRAP solution was mixed with 20 µL diluted samples in a 96-well microplate and incubated in the dark at 37 °C for 30 min. Then, λ = 590 nm wavelength was used to measure the absorbance of samples. Data were indicated as (µM Trolox/g fresh weight (FW).

#### 2.4.3. 2,2′-Azino-bis(3-ethylbenzothiazoline-6-sulfonic acid) Radical Cation-Based Assays (ABTS) Assay

This was undertaken following the previously described method [45]. ABTS assay was briefly performed by adding ABTS solution (7 mM) to potassium persulfate solution (2.4 mM) (1:1 ratio). The mixture was incubated in the dark at 25 °C for 14 h. Diluted samples (1 mL) were mixed with ABTS solution (1 mL) and measured at λ = 734 nm after 7 min. The data were expressed as ABTS inhibition %.

#### 2.4.4. Cupric Ion-Reducing Antioxidant Capacity (CUPRAC) Assay

This was undertaken following the previously assigned method [46]. Concisely, the sample (100 μL) was mixed with 1 mL CuCl_2_ (10 mM), NCP (7.5  mM), and AAB solution (1 M, pH 7.0), and the volume was made up to 4.1 mL with DH_2_O followed by incubation for 30 min, and the samples/absorbance was evaluated at 450 nm. The standard curve was generated using different Trolox concentrations. The data were expressed as (μmol Trolox/g FW).

### 2.5. Untargeted Liquid Chromatography-Mass Spectrum (LC-MS) Metabolome Analysis

A total of 0.2 mL of 80% methanol for each sample was used for LC-MS analysis, as described previously [40] using Q Exactive MS (Thermo, Waltham, MA, USA) and screened with electrospray ionization mass spectrometry (ESI-MS). The LC system comprises an ACQUITY UPLC HSS T3 (100 × 2.1 mm, 1.8 μm) with Ultimate 3000LC. The mobile phase: A (0.05% formic acid-water) and B (acetonitrile) with gradient elution (0–1 min, 5% B; 1–12 min, 5–95% B; 12–13.5 min, 95% B; 13.5–13.6 min, 95–5% B; 13.6–16.0 min, 5% B) with a flow rate of 0.3 mL/min, column temperature of 40 °C, and the sample manager temperature was 4 °C. Mass spectrometry parameters for ESI+ and ESI- mode are listed as follows: heater temp 300 °C; sheath gas flow rate, 45 arbs; aux gas flow rate, 15 arb; sweep gas flow rate, 1 arb; spray voltage, 3.0 kV and 3.2 kV, respectively; capillary Temp, 350 °C; S-Lens RF level, 30% and 60%, respectively.

### 2.6. Nucleic Acid Extraction and qPCR Analysis

The total RNA from the muscadine berry samples, including FS, V, R, and S, was extracted as described previously [47]. Gene-specific primers were designed using Primer Express (v3.0, Applied Biosystems, Waltham, MA, USA) (Appendix A). The real-time quantitative PCR (qPCR) assays were performed as described earlier [47].A CFX384 Touch Real-Time PCR Detection System instrument (BIO-RAD Laboratories) was used with a running system of (95 °C; 5 min, 40 cycles of 95 °C; 10 s, 60 °C; 10 s, and 72 °C; 20 s). Melting curves were created using the following program (95 °C; 15 s, 60 °C; 15 s, and 95 °C; 15 s). Transcript abundance was calculated using standard curves for the target and reference genes and normalized to the reference genes *MrActin* and *MrEF1*.

### 2.7. Metabolite Identification

To obtain metabolite MS/MS spectra, the following databases were used: METLIN [48] (https://metlin.scripps.edu/landing_page.php) (last accessed on 9 February 2022), the Human Metabolome Database [49] (www.hmdb.ca) (last accessed on 11 November 2022), Mass bank [50] (https://massbank.eu/MassBank/Search) (last accessed on 11 November 2022), and MetFrag [51] (https://ipb-halle.github.io/MetFrag/)(last accessed on 30 November 2021) following the standard parameters of metabolite classification [52]. The metabolic pathway map was assembled based on the KEGG (www.kegg.jp/kegg-bin/show) (last accessed on 30 November 2021).

### 2.8. Statistical Analysis

Compound Discover (3.0, Thermo) was used to acquire the raw data based on the *m/z* value and the retention time of the ion signals. ESI- or ESI+ ions were used for multivariate analysis. The principal components analysis (PCA) was first used as an unsupervised method for data visualization and outlier identification. The heat-map analyzed the screened compounds, and statistical analysis was performed by MetaboAnalyst (https://www.metaboanalyst.ca/MetaboAnalyst/home.xhtml) (last accessed on 11 November 2022) and Sigma Plot (version 14.5) to obtain relatively consistent clusters of samples in terms of metabolites.

## 3. Results

### 3.1. Antioxidant Activities

DPPH, FRAP, ABTS, and CUPRAC were determined in the skin (R) and seed (S) of ripe berries of three muscadine genotypes, displaying purple, dark-red, and bronze skin color, using four different antioxidant assays (Figure 1B,C). All the selected genotypes were assessed to detect the prospective differences in their variety of antioxidant activities.

In ripe-skin, the C5 genotype displayed the highest DPPH and FRAP activities estimated at 26.3% ± 3.5 and 469.9 ± 47.0 μM TE/g DW, respectively. The skin of LF and C6 genotypes showed dramatically lower DPPH (30% and 2% of C5, respectively) and FRAP (52% and 42.3% of C5, respectively) activities. Although LF and C6 skin exhibited comparable FRAP activity levels, considerably higher DPPH activity was detected in the LF skin than in C6 (*p* < 0.001). For ABTS activity, the C5 and LF genotypes showed the highest activity with no significant differences (*p* > 0.05). However, the C6 genotype displayed substantially lower activity than C5 (33.0% less activity) and LF (23.1% less activity). Interestingly, the C6 genotype showed significantly the highest CUPRAC antioxidant activity estimated at 265.7 ± 18.5 μmol TE/g DW. The CUPRAC activity in the skin of C5 and LF genotypes reached only 59.9% and 56.6% of those detected in C6, respectively.

Although C5 and LF genotypes demonstrated the highest DPPH and FRAP activities in ripe-seed, no significant differences were detected among the selected genotypes. For ABTS, the C5 genotype revealed the highest activity, but no significant differences with LF seeds were detected (*p* ˃ 0.05). Although the C6 seeds displayed the lowest ABTS activity among the three genotypes, this was significantly accurate with only the C5 seeds (*p* < 0.05). Similar to the CUPRAC activity pattern in ripe-skin, the C6 genotype exhibited the highest activity, estimated at 316.7 ± 22.9 μmol TE/g DW. The CUPRAC activity in LF and C5 seeds represents 82.7% and 78.1% of those that were detected in the C6 genotype.

### 3.2. Metabolite Profiling of Muscadine Skin and Seed Tissues

In muscadine skin and seed tissues, 331 compounds were recognized. They were dispersed as 157, 67, and 107 compounds in positive (ESI+), negative (ESI−), and ESI+/ESI− modes, respectively (Appendix A). Metabolite set enrichment analysis (MSEA) was performed to categorize the identified compounds based on KEGG human metabolic pathways (Figure 2). The bar chart represents the metabolic pathways of the metabolite sets (top 25) under the human metabolic pathway’s library (Figure 2A). Among the top 25 metabolic classes, the primary metabolic pathways of the identified metabolite with a higher *p*-value were mainly arginine biosynthesis, D-glutamine, D-glutamate metabolism, alanine, aspartate metabolism, aminoacyl-tRNA biosynthesis, and citrate cycle (TCA). The node’s color in the metabolic pathways network indicates the different metabolites that were involved in each metabolic pathway (Figure 2B). The interactive pie chart colors indicate individual chemical groups compared to the sum number of compounds (Figure 3A). Among the 12 chemical groups, the organic acids (red) were the highest, followed by fatty acyls (purple), polyketides (blue), and organic heterocycle compounds (green) (Figure 3B).

The PCA analysis approach is to find the differences in the metabolite among the skin and seed tissues in the selected genotypes. The PCA plot (Figure 4A) showed that the first two components explained 95.5% of the variance in the metabolome profile, 87.4% (PC1) and 8.1% (PC2). The PC1 and PC2 were primarily correlated with ripe berry tissues and genotype diversity, respectively. The closest cluster was shown between C5 and LF genotypes indicating the inter-tissue differences in the metabolite profiles. The PCA scatter plot suggested that catechin, gallic acid, and epicatechin-3-gallate were the key metabolites that were present in muscadine seed. However, citramalic acid and malic acid were the main metabolites contributing to muscadine skin (Figure 4B).

### 3.3. Multivariate Analysis of Candidate Metabolites and Antioxidant Activities of Muscadine

PLS-DA was performed to obtain muscadine skin-seed-related metabolites based on VIP scores. As a result, 25 different chemical constituents (VIP > 1) were identified amongst all the metabolites in muscadine seeds and skin (Figure 5). It was observed that the chemical groups of these biomarker compounds in muscadine grapes were highly contributed by organic acids, flavonoids, and phenolic acid. The biomarker compounds in ripe muscadine berries were catechin, gallic acid, citramalic acid, epicatechin-3-gallate, oxoglutaric acid, galactose, pyruvic acid, L-quebrachitol, quercetin 4′-glucoside, ellagic acid, malic acid, succinic acid, fertaric acid, hydroxy propionic acid, quinic acid, deoxyribose, procyanidin B1, oxalic acid, malonic acid, glucobrassicin, 7-methylxanthine, tartaric acid, 2-furoic acid, D-xylose, and pyroglutamic acid. A previous study stated that phenolic acid derivatives, flavonoids, gallic acid, ellagic acid derivatives, and tannins mainly contributed to muscadine grape phenolics [53]. To better understand the differences in metabolites and antioxidant activities, those 25 candidate compounds and DPPH, FRAP, ABTS, and CUPRAC of muscadine grapes were plotted with a heatmap. In Figure 6, each column represents the muscadine grape genotypes in skin and seed tissues, and each line represents the metabolites and antioxidant activities of three replicates. The yellow and blue color in the plot represents higher and lower metabolite intensities and antioxidant activities. By comparing the color intensity variation across all samples, it was observed that the highest DPPH and FRAP antioxidant activities were detected in the S tissue for all genotypes. By contrast, CUPRAC was highest in the seed of only the C6 genotype. However, the ABTS activity was higher in C5-S and LF-S. Furthermore, in the seed of all genotypes, catechin, epicatechin-3-gallate, and fertaric acid were more abundant. The procyanidin B1 was dominant in C5-S and C6-S, whereas gallic acid was maximum in the LF and C5 genotypes seed. Succinic and ellagic acids were prevalent in LF-S and C5-S, respectively.

The heatmap Pearson’s correlation coefficient analysis was accomplished to evaluate the correlation between marker compounds and different antioxidant activities (Figure 7), cluster analysis suggested that DPPH and FRAP antioxidant activities had a comparable correlation pattern rather than ABTS and CUPRAC activities which are slightly correlated with DPPH and FRAP activities. This assessment conforms with the antioxidant results in Figure 1B,C. A total of five key compounds were highly correlated to DPPH and FRAP antioxidant activities and slightly correlated with ABTS and CUPRAC antioxidant activities, including catechin, gallic acid, epicatechin-3-gallate, fertaric acid, and procyanidin B1. Interestingly, the five biomarker compounds were highly accumulated in the seed tissue relative to the skin (Figure 8). Among the five biomarker compounds, the quantities of catechin, epicatechin-3-gallate, and procyanidin B1 in the skin tissue represent only 0.7% ± 0.3 of those that were detected in the seed. However, gallic acid and fertaric acid in the skin denote 3.5% ± 0.6 of their quantity in the seed.

### 3.4. Expression of Genes Underlying Biomarker Compounds Synthesis

The metabolome data and the transcription of 15 crucial candidate genes underlying the biomarker compounds accumulation and antioxidant activity were assessed during berry development of the C5 genotype using the qPCR approach (Figure 9, Appendix A).

Analysis of the expression data during development differentiated the transcripts based on their accumulation pattern into four main groups. Group-I includes all mRNAs that manifested the highest levels at the early FS stage despite their significant accumulation in seed that is represented by the 3-dehydroquinate dehydratase (*DHD*), shikimate kinase (*SK*), chalcone synthase (*CHS*), anthocyanidin reductase (*ANR*), and laccase (*LAC*). Group-II comprises transcripts that were similarly accumulated in high levels at the FS stage and seed tissue, including the phenylalanine ammonia-lyase (*PAL*) and dihydroflavonol 4-reductase (*DFR*). Group-III holds all transcripts that demonstrated the highest levels in seed despite their abundance at the FS stage, which is represented by 3-dehydroquinate synthase (*DHQS*), chorismate mutase (*CM*), and flavanone-3-hydroxylase (*F3H*). Finally, Group-IV covers the different mRNAs exclusively accumulated in seed, including the cinnamoyl-CoA reductase (*CCR*), cinnamyl alcohol dehydrogenase (*CAD*), leucoanthocyanidin reductase (*LAR*), gallate 1-β-glucosyltransferase (*UGT*), and anthocyanidin 3-O-glucosyltransferase (*UFGT*).

Gene expression and metabolomics analysis can comprehensively elucidate the accumulation profile of antioxidant-related metabolites in muscadine ripe berry tissues, seed and skin. For instance, the genes encoding enzymes contributed to the gallic pathway (Figure 10), including the *DHQS* that converts the 3-deoxy-D-arabinoheptulosonic acid 7-phosphate to 3-dehydroquinic acid and the *DHD* that converts the dehydroshikimic acid to gallic acid were highly abundant in muscadine seed relative to skin.

Similarly, for the fertaric acid pathway (Figure 10), the high accumulation of fertaric acid in muscadine ripe-seed was associated with the abundance of several gene-encoding enzymes, including the *SK*, *CM*, *PAL*, *CCR*, and *CAD* mRNAs. Finally, the high levels of catechin, epicatechin-3-gallate, and procyanidin B1 were allied with the accumulation of *CHS*, *F3H*, *DFR*, *LAR*, *ANR*, *UGT*, *UFGT*, and *LAC* transcripts in muscadine ripe-seed tissue.

## 4. Discussion

The antioxidant activities of plant extracts are multifunctional and dependent on the structure-activity relationship of the bioactive metabolite compositions. Therefore, characterizing the antioxidant activities using different techniques would provide various aspects of their antioxidant capacities [54].

Typically, grape seed tissue accumulates more bioactive compounds, predominantly flavan-3-ols and proanthocyanidins [40,41]. To determine the quality of antioxidant activity in ripe-skin, we calculated the ratio of activity in skin relative to seed. Although skin showed significantly lower activities than seed, the skin/seed ratio was dependent on the type of antioxidant activity. DPPH and FRAP average activities were much higher in seed, resulting in relatively low ratios. The average DPPH and FRAP activities in the skin represent only 12.1% ± 13.7 and 6.7% ± 3.3 of those that were detected in the seed. However, ABTS and CUPRAC activities in the skin were relatively high, resulting in high skin/seed ratios. The average ABTS and CUPRAC activities in the skin represent 79.3% ± 9.5 and 68.8% ± 13.8 of those detected in seeds.

Analysis of antioxidant capacities suggested that the activity occurred in a genotype-dependent manner and irrespective of berry color. Among the evaluated assays, the dark red skin C6 genotype showed considerably higher CUPRAC activity. Conversely, the bronze skin LF genotype displayed significantly higher DPPH and ABTS activities. However, both genotypes exhibited comparable FRAP activity. Recently published data demonstrated that the absence of anthocyanin pigments from muscadine berries does not influence their nutraceutical qualities, as such genotypes accumulate more proanthocyanidins in their skin [40,55].

The primary metabolic pathways of the identified metabolite based on MSEA analysis were mainly arginine biosynthesis, D-glutamine, and D-glutamate metabolism. Among the 12 chemical groups, the organic acids were the highest, followed by fatty acyls. Other studies suggested that organic acids, carbohydrates, and phenolics represent the major chemical components in grapes [13].

Similar to our findings, another study reported that the capability of catechins for antioxidant activity is applied through different mechanisms such as ROS scavenging, chelating metal ions, inducing antioxidant enzymes, inhibiting pro-oxidant enzymes, and producing phase II detoxification enzymes and antioxidant enzymes [56]. Another antioxidant potential for epicatechin-3-gallate has been explained as a potent scavenger of a wide range of ROS and an effective LPO inhibitor [57]. Catechin and epicatechin gallate were most efficient in inhibiting AAPH-induced oxidation of 2′7′-dichlorodihydrofluorescein contained inside erythrocytes [58]. Gallic acid (3,4,5-trihydroxy benzoic acid) is the main phenolic compound that is found in many natural sources with high antioxidant activity [59]. The antioxidant principle of gallic acid possibly involves the hydrogen-donating mechanism. This also could be due to the hydroxyl groups arrangement on the aromatic ring [60]. Proanthocyanidin B1 is pervasive in fruits and has a potent scavenger of oxygen free radicals, and targets harmful signaling pathways activated downstream of free radical production [61]. Moreover, it prevents OS-induced DNA damage and promotes DNA repair through different pathways, such as scavenging oxidative species and free radicals [62]. One more antioxidant determinant is Fertaric acid, which modulates Nrf2 antioxidant and inflammatory pathways [63]. Fertaric acid (FA) is also a hydroxycinnamic acid found in grapefruit, and its antioxidant activity was determined using in silico techniques [64]. Exceptional antioxidant properties of the identified biomarker compounds make them ideal candidates for nano-formulations to be used in antioxidant therapy.

Polyphenols have antioxidant powers and are produced to protect plants from oxygen-reactive species [65]. These compounds have antioxidant properties and contribute to the pharma-therapeutic properties of food products that are derived from plants [66]. Therefore, the representation of polyphenols as antioxidant agents is essential in human nourishment. Flavonoids and other plant polyphenols retain free radical scavenging properties. Some publications reported the strong association between the chemical structure of metabolites and activity, DPPH and FRAP properties of pure compounds, and established connections between flavonoids and high DPPH or FRAP activities. In flavonoids, OH groups on ring-B positively affected DPPH and FRAP activities. A 3-OH on ring-C also increased the FRAP but did not affect DPPH activity. Conversely, phenolic and hydrocinnamic acids lacking a 3-OH had significantly lower FRAP or DPPH activities than compounds with this structure [67]. These data strongly support our findings that the DPPH and FRAP activities of the seed extract were higher than skin extract due to the high accumulation of flavonoids (catechin, epicatechin-3-gallate, and procyanidin B1) rather than phenolic and hydrocinnamic acids (gallic and fertaric acids). The differences in antioxidant activity between skin and seed tissues correlated to the accumulation of genes encoding proteins that are involved in the biomarker compounds synthesis. Recently published data suggested that early berry developmental stages of muscadine grapes exhibited comparable antioxidant activities to those detected in ripe-seed tissue [40].

## 5. Conclusions

In summary, we investigated the variations in antioxidant activity and untargeted metabolite profiles among three muscadine cultivars in ripe-skin and -seed tissues. The untargeted metabolite profiling revealed that the chemical composition of muscadine extracts is dominated by organic acids, fatty acyls, polyketides, and organic heterocycle compounds. We also demonstrated that muscadine ripe-seeds extract had great potential for antioxidant activity using four different methods DPPH, FRAP, ABTS, and CUPRAC. Using the multivariate statistical analysis, we identified five muscadine biomarker metabolites, catechin, gallic acid, epicatechin-3-gallate, fertaric acid, and procyanidin B1, that exhibited higher positive correlations with antioxidant activities. The five biomarker compounds were highly accumulated in the ripe-seeds relative to the ripe-skin. Even though the nutritional value of muscadine berries is known, no previous research has been performed to explain the molecular basis underlying potent berry bioactivity. The current study provides gene expression profiling to explore genetic markers and QTLs associated with specific metabolite content and antioxidant activity. The evaluated transcripts are extremely helpful for investigating antioxidant-related genes that become strongly expressed and co-regulated during berry development to provide significant insight into time-spatial dynamics emphasizing differences in gene expression that are associated with antioxidant activity. We identified 15 genes that were strongly associated with biomarker biosynthesis. The informed database can be used for future phenotypic-genotypic association studies. Our findings suggested that muscadine grape seeds contain essential metabolites that could attract the attention of those that are interested in the pharmaceutical sector and the plant breeders to develop new varieties with high nutraceutical values.

## Figures and Tables

**Figure 1 metabolites-13-00210-f001:**
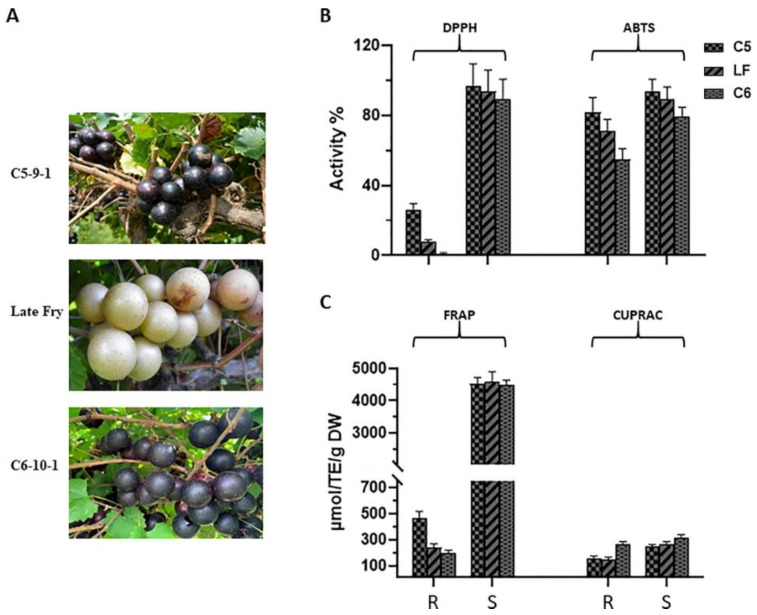
(**A**) A representative image of muscadine grape genotypes C5, Latefry, and C6. DPPH and ABTS (**B**), FRAP, and CUPRAC (**C**) activities. During ripening (R), and seed (S) stages, antioxidant activities of muscadine grape genotypes were determined using DPPH and ABTS (**B**), FRAP, and CUPRAC (**C**) assays. The experiments were carried out in three biological replicates, and each replicate was repeated three times (*n* = 9). The data represent the mean values ± SD (*n* = 3).

**Figure 2 metabolites-13-00210-f002:**
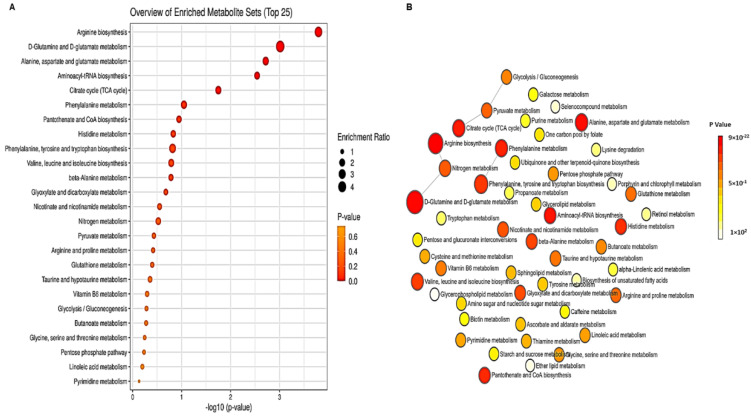
(**A**) Bar chart and (**B**) interactive network chart metabolite set enrichment analysis (MSEA) were conducted to classify the identified compounds in ripe muscadine berries based on KEGG human metabolic pathways. Colors in the bar plot describe the *p*-value. The red and orange colors signify the high and low values, respectively. The lines indicate the enrichment ratio computed by hits/expected, where hits = observed hits and expected = expected hits. The red, orange, and yellow colors in the interactive network chart designate each metabolic pathway relative to the total number of compounds.

**Figure 3 metabolites-13-00210-f003:**
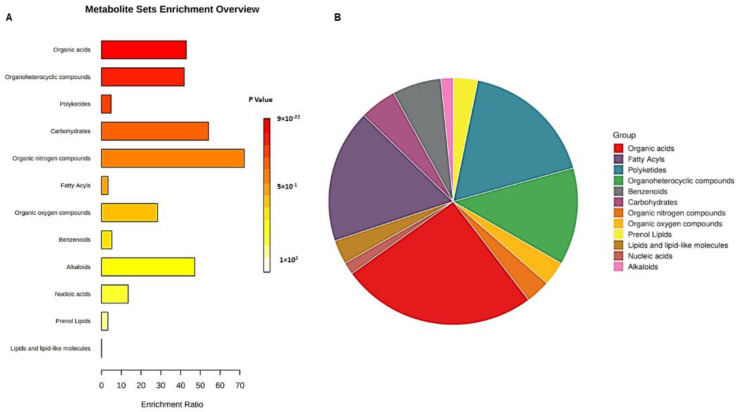
(**A**) Bar chart and (**B**) interactive pie chart of the chemical classification of ripe muscadine berry metabolites in skin and seed tissues using metabolite set enrichment analysis (MSEA). Colors in the bar plot describe the *p*-value. The red and orange colors signify the high and low values, respectively. The lines indicate the enrichment ratio, which was computed by hits/expected, where hits = observed hits and expected = expected hits. The colors in the interactive pie chart designate each chemical group relative to the total number of compounds.

**Figure 4 metabolites-13-00210-f004:**
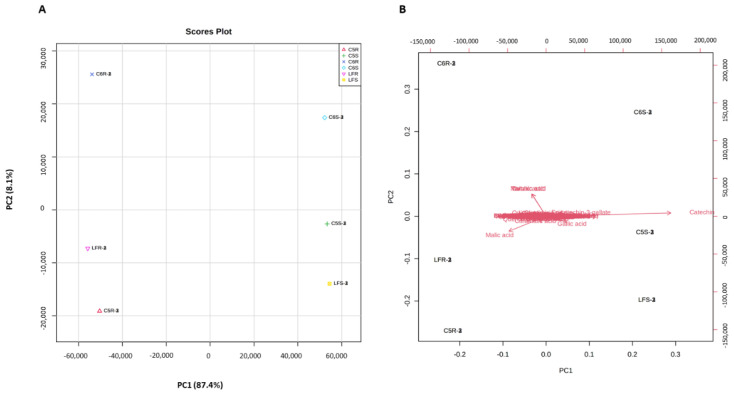
PCA 2D score plot (**A**) and biplot (**B**) of muscadine skin and seeds metabolites. The muscadine genotypes and stages of development demonstrated by C5R, C5S, C6R, C6S, LFR, and LFS. The red (Δ) represents C5-R, green (+) represents C5-S, purple (×) represents C6-R, blue (◊) represents C6-S, pink (Δ) LF-R, and the yellow (**□**) represents LF-S.

**Figure 5 metabolites-13-00210-f005:**
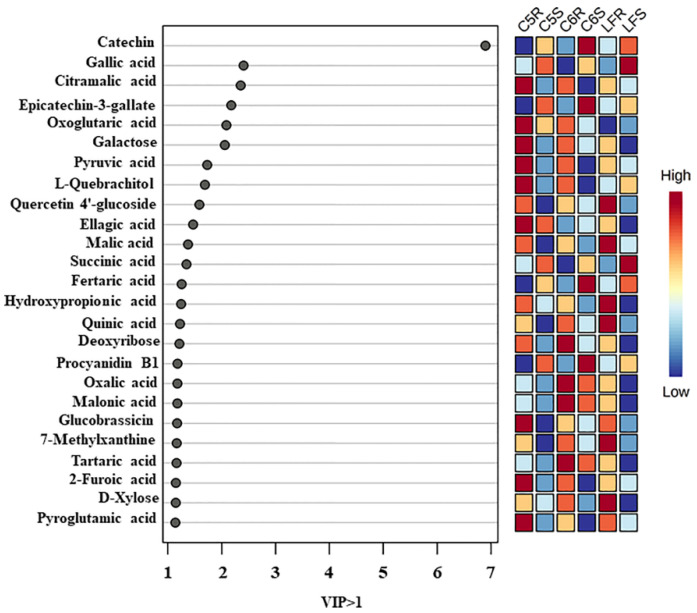
The weighted sum of absolute regression coefficients (coef.) of candidate metabolites (VIP > 1) of ripe muscadine skin and seed. The colored boxes on the right indicate the relative concentrations of the corresponding metabolite in each group under study.

**Figure 6 metabolites-13-00210-f006:**
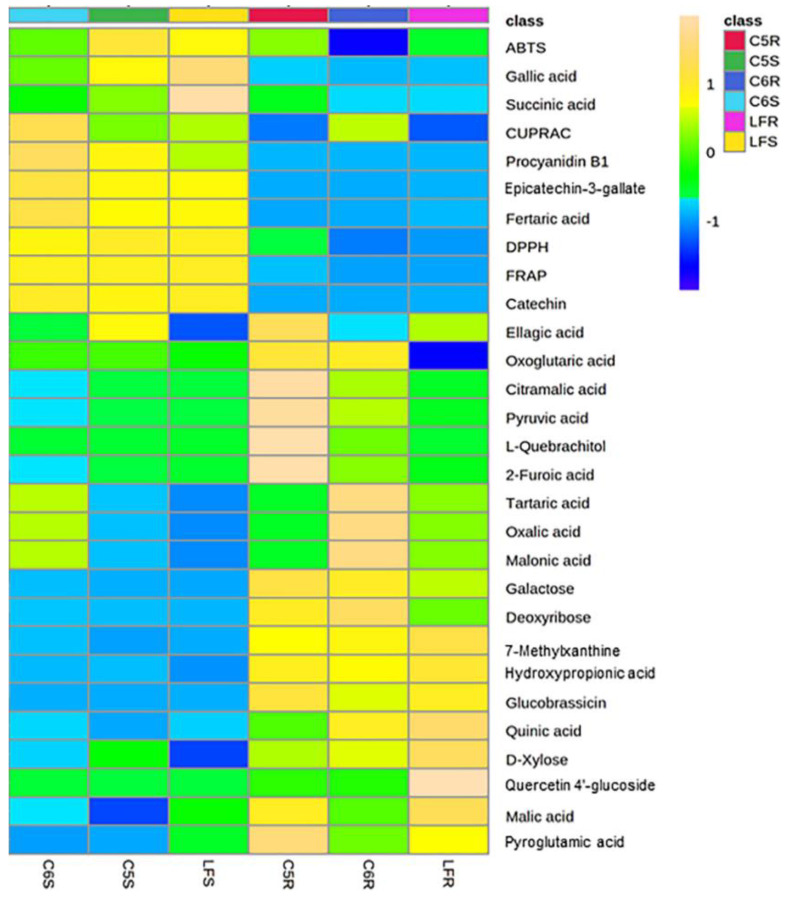
Heatmap analysis of candidate metabolites (VIP > 1) that were obtained by partial least-squares–discriminant analysis (PLS-DA) and antioxidant activities of ripe muscadine berries. Each column refers to the muscadine genotype with seed or skin tissues, and each row indicates the metabolites and antioxidant activities. The yellow and blue colors in the plot describe high and low intensities, and the values range from −2 to +2. The higher yellow color intensity (from +1 to +2 values) represents the greater metabolite contents and antioxidant activities. By contrast, the higher blue color intensity (from −1 to −2 values) describes the lower metabolite contents and antioxidant activities.

**Figure 7 metabolites-13-00210-f007:**
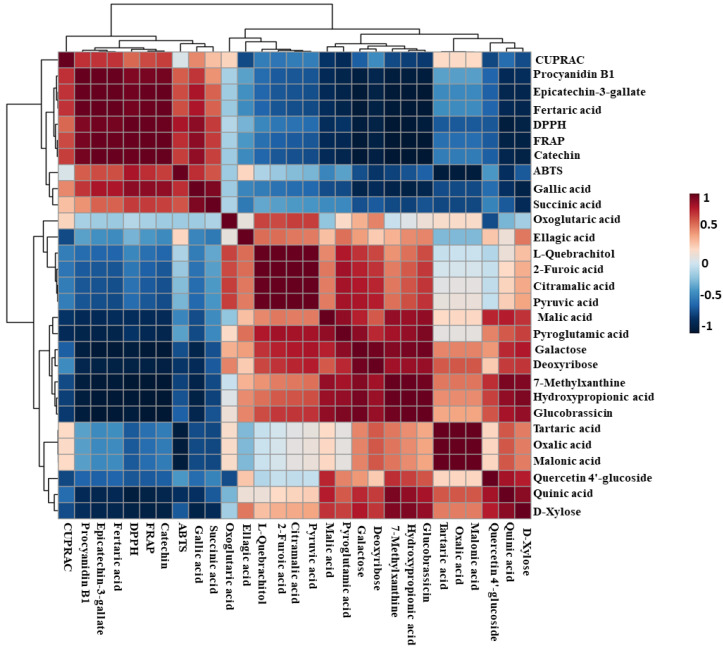
Heatmap of Pearson correlation between candidate metabolites (VIP > 0.4) with antioxidant activities of muscadine genotypes at selected developmental stages. Correlation values range from −1 to +1. The values close to +1 represent the higher positive correlation, whereas values closer to zero mean no linear trend between the variables; values close to −1 represent the negative correlation between variables.

**Figure 8 metabolites-13-00210-f008:**
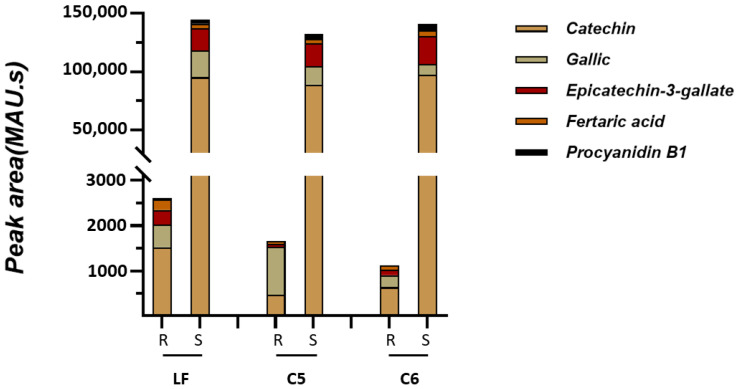
The intensity of absorbance (MAU.s) values of muscadine ripe and seed biomarker compounds among different genotypes.

**Figure 9 metabolites-13-00210-f009:**
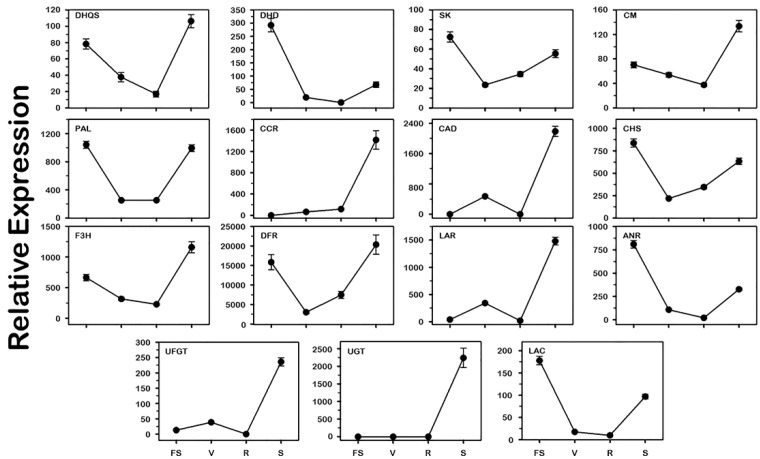
Expression profile of the 15 selected muscadine genes quantified by qPCR related to antioxidant activity in the C5 genotype at different stages of berry development. The genes encode *DHQS* 3-dehydroquinate synthase, *DHD* 3-dehydroquinate dehydratase, *SK* shikimate kinase, *CM* chorismate mutase, *PAL* phenylalanine ammonia-lyase, *CCR* cinnamoyl-CoA reductase, *CAD* cinnamyl alcohol dehydrogenase, *CHS* chalcone synthase, *F3H* flavanone-3-hydroxylase, *DFR* dihydroflavonol 4-reductase, *LAR* leucoanthocyanidin reductase, *ANR* anthocyanidin reductase, *UGT* gallate 1-β-glucosyltransferase, *UFGT* anthocyanidin 3-O-glucosyltransferase, and *LAC* laccase. The *y*-axis represents the mean expression level (±SD) determined by qPCR. The *x*-axis in each chart represents the developmental stages (fruit-set FS, véraison V, ripe-skin R, and ripe-seed S). The level of expression mean was calculated from three bio/tech replicates (*n* = 9). Standard curves were used to calculate the number of target gene molecules/samples and standardized relative to the expression of *actin* and *EF1*.

**Figure 10 metabolites-13-00210-f010:**
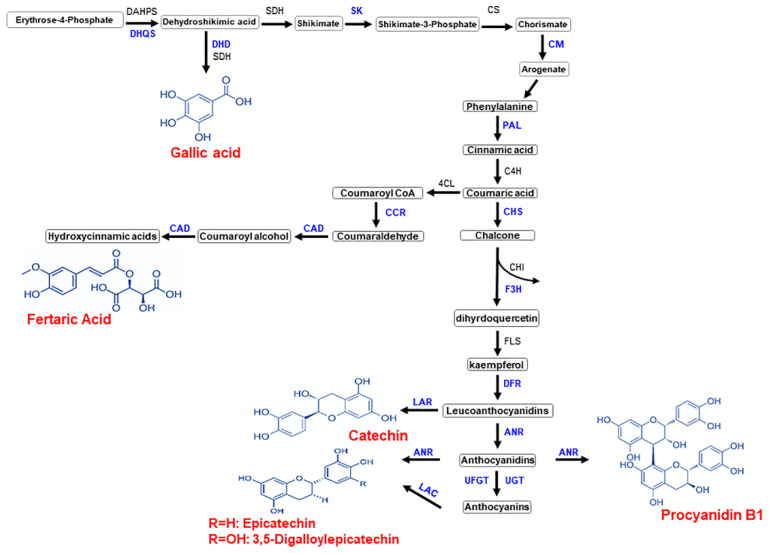
The pathway shows the different genes involved in biomarker compounds biosynthesis pathways. *DHQS* 3-dehydroquinate synthase, *DHD* 3-dehydroquinate dehydratase, *SK* shikimate kinase, *CM* chorismate mutase, *PAL* phenylalanine ammonia-lyase, *CCR* cinnamoyl-CoA reductase, *CAD* cinnamyl alcohol dehydrogenase, *CHS* chalcone synthase, *F3H* flavonoid 3′- hydroxylase, *DFR* dihydroflavonol 4-reductase, *LAR* leucoanthocyanidin reductase, *ANR* anthocyanidin reductase, *UGT* UDP-glycosyltransferase, *UFGT* UDP-glucose: flavonoid 3-O-glucosyltransferase, and *LAC* laccase.

## Data Availability

The data presented in this study are available on request from the corresponding author. The data are not publicly available due to privacy.

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
