# Peer review of "Integrating Metabolomics and Gene Expression Underlying Potential Biomarkers Compounds Associated with Antioxidant Activity in Southern Grape Seeds"

_metabolites, 2023, doi:10.3390/metabo13020210_

Round 1
Reviewer 1 Report
Reviewer report for paper entitled "Integrating metabolomics and gene expression underlying tential biomarkers compounds associated with antioxidant activity in southern grape seeds". The manuscript is well written and prepared in very good scientific sequence. The antioxidant properties of grape seeds is well-known since long time ago. However, this work provide in depth more in depth information to understand the antioxidant properties in molecular level. The work is considered novel and of high interest. However, some improvements need to be done before publishing this work as follows:
- In materials and methods part. Need to provide more information about the sample used in this study, the latine name (as in abstract part) and more information about the different varieties used in this study as well as the specific location of the experiment in GPS, time of year and season.
- Need to provide more information about the conditions of the Muscadine extracts. In addition, for all equipment used need to provide (Model, Company, City, Country).
- The metabolic profiling study is well done and provided some new information about the bioactive compounds of antioxidant activities.
- The biomarker compounds biosynthesis experiment is well done.
- The Conclusion part need to address the potential application of the obtained results and the future perspectives and recommendation as well.
Author Response
We appreciate you and the reviewers for your precious time in reviewing our paper and providing valuable comments. Your valuable and insightful comments led to possible improvements in our manuscript entitled “Integrating metabolomics and gene expression underlying potential biomarkers compounds associated with antioxidant activity in southern grape seeds.”
The authors have carefully considered the comments and tried our best to address every one of them. We hope the manuscript, after careful revisions, meets your high standards. The authors welcome further constructive comments, if any.
Below we provide the point-by-point responses. All modifications in the manuscript have been marked up using “Track Changes”.
Reviewer #1:
- In materials and methods part. Need to provide more information about the sample used in this study, the Latin name (as in the abstract part), and more information about the different varieties used in this study as well as the specific location of the experiment in GPS, time of year, and season.
All needed information has been added.
- Need to provide more information about the conditions of the Muscadine extracts. In addition, all equipment used needs to provide (Model, Company, City, Country).
The authors have added all the required information related to muscadine extracts (muscadine extraction, equipment information, etc.) as per the reviewer’s suggestion.
- The metabolic profiling study is well done and provided some new information about the bioactive compounds of antioxidant activities.
The authors appreciate the reviewer’s comment.
- The biomarker compounds biosynthesis experiment is well done.
The authors appreciate the reviewer’s comment.
- The Conclusion part needs to address the potential application of the obtained results and the future perspectives and recommendations as well.
The obtained results have been addressed well in the conclusion part. Also, future perspectives and recommendation has been included.

Reviewer 2 Report
1. It is recommended that the abstract be supplemented with the main thrust of the article's research.
2. The article states websites or software for heatmaps and PCA charts and suggests additional software or websites that do other charts.
3. The results of the composition and antioxidant activity to be expressed in the heat map of Figure 7 are not intuitive and it is recommended that the heat map be modified or that the presentation of the results be further refined.
4. It is suggested that the discussion section be supplemented with additional components and pathways of action for antioxidant activity.
5. It is recommended that the logic of the conclusion section be sorted out and the description of the conclusion section refined.
6. The "A" and "B" in Fig. 1 are incomplete; Fig. 3 has the word "251" behind the top left corner; Fig. 5 has the word "226" behind it, etc. It is recommended that all images in the article be checked and adjusted.
Author Response
We appreciate you and the reviewers for your precious time in reviewing our paper and providing valuable comments. Your valuable and insightful comments led to possible improvements in our manuscript entitled “Integrating metabolomics and gene expression underlying potential biomarkers compounds associated with antioxidant activity in southern grape seeds.”
The authors have carefully considered the comments and tried our best to address every one of them. We hope the manuscript, after careful revisions, meets your high standards. The authors welcome further constructive comments, if any.
Below we provide the point-by-point responses. All modifications in the manuscript have been marked up using “Track Changes”.
Reviewer #2:
- It is recommended that the abstract be supplemented with the main thrust of the article's research.
The abstract has been modified as per the reviewer’s comment.
- The article states websites or software for heatmaps and PCA charts and suggests additional software or websites that do other charts.
The authors confirm that only the MetaboAnalyst (https://www.metaboanalyst.ca/MetaboAnalyst/home.xhtml) and Sigma Plot have been used to do the heatmaps, PCA, and other analysis.
- The results of the composition and antioxidant activity to be expressed in the heat map of Figure 7 are not intuitive and it is recommended that the heat map be modified or that the presentation of the results be further refined.
The heat map has been modified. Also, the results have been addressed carefully to explain the same data.
- It is suggested that the discussion section be supplemented with additional components and pathways of action for antioxidant activity
Additional information related to the mechanism of action for antioxidant activity has been added to the discussion section.
- It is recommended that the logic of the conclusion section be sorted out and the description of the conclusion section refined.
The conclusion section has been sorted logically and refined as per the reviewer’s comment.
- The "A" and "B" in Fig. 1 are incomplete; Fig. 3 has the word "251" behind the top left corner; Fig. 5 has the word "226" behind it, etc. It is recommended that all images in the article be checked and adjusted.
All figures have been modified based on the reviewer’s comments.
Reviewer 3 Report
Reviewer Comments
General Comment:
The manuscript is well written. The authors investigated “Integrating Metabolomics and Gene Expression Underlying Potential Biomarkers Compounds Associated with Antioxidant Activity in Southern Grape Seeds”. The study is interesting and adds to the existing body of knowledge. Some errors need clarification and revisions. The manuscript needs to undergo English editing.
Detail’s comment:
1. Page 1, Line 37-40: Please more citations regarding this statement.
2. Page 2, Line 82-88: Please add the figure of the muscadine grape in this section. Please separated it from Figure 1 in section 3.1.
3. Page 2, Line 90: Please state the exact weight of the skin and powder sample of muscadine. Please also state the extraction ratio for the sample and solvent. Also, please state the extraction yield for this section.
4. Page 3, Line 99: Please write the full name for DPPH and then abbreviated it.
5. Page 3, Line 101: Please write the concentrations in increasing dose (3.12-100 ug/ml)
6. Page 3, Line 108: Please write the full name for FRAP and then abbreviated it.
7. Page 3, Line 111: absorbance. No need to abbreviate.
8. Page 3, Line 113: Please write the full name for ABTS and then abbreviated it. Lines 116 and 117: 1 mL
9. Page 3, Line 199: Please write the full name for CUPRAC and then abbreviated it.
10. Page 3, Line 126-128: Please write the full name for LC-MS, and ESI-MS and then abbreviated it.
11. Page 4, Line 169: Figures 1B and 1C. Not Fig. 1
12. Page 4, Line 172: Figure legend (A) Image of the ripening muscadine grape genotypes Latefry, C5, and C6. But the figure showed C5-9-1, LateFry, and C6-10-1. Please check back.
13. Page 4, Figure 1B and 1C: Any significant difference in antioxidant activities?
14. Page 5, Line 176: 26.3 ± 3.5% and 469.9 ± 47.0%. Lines 175-185: Any previous studies reported similar current study findings in DPPH, FRAP, and CUPRAC?
15. Page 5, Line 179-180: p = 0.00093 or p < 0.001
16. Page 5, Line 181: Please remove P = 0.159 or put P > 0.05.
17. Page 5, Line 189: Please remove P = 0.471 or put P > 0.05.
18. Page 5, Line 216: Supplementary Table 1.
19. Page 10, Line 426: Figure 7, please replace with the high-resolution figure.
20. Page 12, Line 540: Figure 10 legend: Please write the full name and abbreviation for all the genes.
Author Response
We appreciate you and the reviewers for your precious time in reviewing our paper and providing valuable comments. Your valuable and insightful comments led to possible improvements in our manuscript entitled “Integrating metabolomics and gene expression underlying potential biomarkers compounds associated with antioxidant activity in southern grape seeds.”
The authors have carefully considered the comments and tried our best to address every one of them. We hope the manuscript, after careful revisions, meets your high standards. The authors welcome further constructive comments, if any.
Below we provide the point-by-point responses. All modifications in the manuscript have been marked up using “Track Changes”.
Reviewer #3:
- The manuscript needs to undergo English editing.
The manuscript has been edited and revised by a native English-speaking colleague.
- Page 1, Line 37-40: Please more citations regarding this statement.
More citations have been added to sufficiently cover this part.
- Page 2, Line 82-88: Please add the figure of the muscadine grape in this section. Please separated it from Figure 1 in section 3.1.
We appreciate the reviewer’s comment. However, the authors preferred to keep the figure as is to respect the number of figures in the manuscript.
- Page 2, Line 90: Please state the exact weight of the skin and powder sample of muscadine. Please also state the extraction ratio for the sample and solvent. Also, please state the extraction yield for this section.
The exact sample weight, the extraction ratio for the sample and solvent, and the extraction yield have been added.
- Page 3, Line 99: Please write the full name for DPPH and then abbreviated it.
The DPPH full name has been added.
- Page 3, Line 101: Please write the concentrations in increasing dose (3.12-100 ug/ml).
The authors have modified the writing of the concentrations to be in order from the lowest to the highest concentration.
- Page 3, Line 108: Please write the full name for FRAP and then abbreviated it.
The FRAP full name has been added.
- Page 3, Line 111: absorbance. No need to abbreviate.
The full name has been added.
- Page 3, Line 113: Please write the full name for ABTS and then abbreviated it. Lines 116 and 117: 1 mL
The full name of the ABTS assay has been added, and we modified ml to mL
- Page 3, Line 199: Please write the full name for CUPRAC and then abbreviated it.
The full name of the CUPRAC assay has been added.
- Page 3, Line 126-128: Please write the full name for LC-MS, and ESI-MS and then abbreviated it.
LC-MS and ESI-MS full names have been added.
- Page 4, Line 169: Figures 1B and 1C. Not Fig. 1
Fig.1 has been modified to be figures 1B and 1C.
- Page 4, Line 172: Figure legend (A) Image of the ripening muscadine grape genotypes Latefry, C5, and C6. But the figure showed C5-9-1, LateFry, and C6-10-1. Please check back.
The figure legend has been modified to show the genotypes in order, as shown in the figure.
- Page 4, Figure 1B and 1C: Any significant difference in antioxidant activities?
The authors thank the reviewer for his comment. However, as we mentioned in lines 216-246. A significant difference exists between the genotypes regarding antioxidant assay and the grape tissue used (Skin/Seed).
- Page 5, Line 176: 26.3 ± 3.5% and 469.9 ± 47.0%. Lines 175-185: Any previous studies reported similar current study findings in DPPH, FRAP, and CUPRAC?
Yes, there are some published manuscripts, including our previous published paper showed similar findings to our current study https://doi.org/10.3390/antiox10060914
- Page 5, Line 179-180: p = 0.00093 or p < 0.001
The statement has been modified as per the reviewer’s comment.
- Page 5, Line 181: Please remove P = 0.159 or put P > 0.05.
The statement has been modified as per the reviewer’s comment.
- Page 5, Line 189: Please remove P = 0.471 or put P > 0.05.
The statement has been modified as per the reviewer’s comment.
- Page 5, Line 216: Supplementary Table 1.
The statement has been modified as per the reviewer’s comment.
- Page 10, Line 426: Figure 7, please replace with the high-resolution figure.
We replaced the figure with a high-resolution one.
- Page 12, Line 540: Figure 10 legend: Please write the full name and abbreviation for all the genes.
We modified the figure 10 legend, including the full name of genes.

Reviewer 4 Report
The present work intended to investigate the metabolic profiles and antioxidant ac-68 tivities of phytochemicals accumulated in muscadine ripe berries, which could be categorized as biomarker compounds with antioxidant potential. Findings demonstrate that muscadine grape seeds contain essential metabolites that could attract the attention of whom are interested in the pharmaceutical sector.
Results obtained are well explained and data interpretation is also correct. Conclusions are consistent with the evidence and arguments presented. The methods used are sufficiently documented and allow replication studies.
About strenghts the authors explored the topic and they obtained the purpose of the study.
However, I will recommend the acceptance of this manuscript after these modifications:
-About references 3, 13, 19-27, 33, 56 the date of their publication is too old and authors should modify it;
-The reported acronymus should be explain in extenso at their first appearance in the text: DW, DPPH, FRAP, CUPRAC, ABTS, NO, NADPH, DPTZ, SNP, PMS, NCP.
Author Response
We appreciate you and the reviewers for your precious time in reviewing our paper and providing valuable comments. Your valuable and insightful comments led to possible improvements in our manuscript entitled “Integrating metabolomics and gene expression underlying potential biomarkers compounds associated with antioxidant activity in southern grape seeds.”
The authors have carefully considered the comments and tried our best to address every one of them. We hope the manuscript, after careful revisions, meets your high standards. The authors welcome further constructive comments, if any.
Below we provide the point-by-point responses. All modifications in the manuscript have been marked up using “Track Changes”.
Reviewer #4:
- About references 3, 13, 19-27, 33, and 56, the date of their publication is too old, and authors should modify it.
We replaced the old references with recent references as much as we could.
- The reported acronymous should be explained in extenso at their first appearance in the text: DW, DPPH, FRAP, CUPRAC, ABTS, NO, NADPH, DPTZ, SNP, PMS, NCP.
All abbreviations have been reported in a full name in the manuscript.

Round 2
Reviewer 2 Report
1. It is suggested to look at the article structure required by the journal and divide "Results and Discussion" into "Results" and "Discussion".
2. After dividing "Results and Discussion" into "Results" and "Discussion", the research objects and results are discussed in the "Discussion" section, and key contents such as the mechanism of action are indicated.
3. It is suggested to summarize the research Conclusion in "4. conclusion", eliminating redundant text description.
Author Response
Reviewer #2:
- It is suggested to look at the article structure required by the journal and divide "Results and Discussion" into "Results" and "Discussion".
The results and discussion section have been separated as per journal format.
- After dividing "Results and Discussion" into "Results" and "Discussion", the research objects and results are discussed in the "Discussion" section, and key contents such as the mechanism of action are indicated.
The required modification has been done as per the reviewer’s suggestion.
- It is suggested to summarize the research Conclusion in "4. conclusion", eliminating redundant text description.
The conclusion section has been revised and edited.
Sincerely,
Ahmed G. Darwish